# HERVH-derived lncRNAs negatively regulate chromatin targeting and remodeling mediated by CHD7

Fu-Kai Hsieh[1,2], Fei Ji[1,2], Manashree Damle[1,2], Ruslan I Sadreyev[1,3], Robert E Kingston[1,2]

*Chd7* **encodes an ATP-dependent chromatin remodeler which has been shown to target specific genomic loci and alter local transcription potentially by remodeling chromatin structure. De novo mutations in CHD7 are the major cause of CHARGE syndrome which features multiple developmental defects. We examined whether nuclear RNAs might contribute to its targeting and function and identified a preferential interaction between CHD7 and lncRNAs derived from HERVH loci in pluripotent stem cells. Knockdown of HERVH family lncRNAs using LNAs or knockout of an individual copy of HERVH by CRISPR-Cas9 both resulted in increased binding of CHD7 and increased levels of H3K27ac at a subset of enhancers. Depletion of HERVH family RNAs led to the activation of multiple genes. CHD7 bound HERVH RNA with high affinity but low specificity and this interaction decreased the ability of CHD7 to bind and remodel nucleosomes. We present a model in which HERVH lncRNAs act as a decoy to modulate the dynamics of CHD7 binding to enhancers in pluripotent cells and the activation of numerous genes that might impact the differentiation process.**

## Introduction

Proper development requires both the establishment of lineage-specific gene expression patterns and the maintenance of those patterns. Numerous proteins and complexes required for these processes have been characterized, many of which are encoded by members of the Polycomb-Group and trithorax-Group (trxG), two families of genes initially identified in Drosophila genetic screens (Mills, 2010). Proteins in these complexes frequently function by modifying chromatin, either via covalent modification of histones or by increasing or decreasing the ability of nucleosomes to move fluidly on genes and gene regulatory elements. Nucleosomes, the essential packaging element of eukaryotic DNA, consist of 147 bp DNA wrapped 1.7 turns around a histone octamer composed of two copies each of histones H2A, H2B, H3, and H4 (Luger et al, 1997). The

long "beads on a string" structure of nucleosomal DNA can be further organized into higher order chromatin structure and formed the accessible euchromatin or the tightly condensed heterochromatin. These various levels of organization need to be properly controlled to allow the embedded DNA sequences to be bound by regulatory factors that modulate appropriate transcription, replication and DNA repair (Maeshima et al, 2019).

ATP-dependent chromatin remodeling is a prominent mechanism to modulate chromatin dynamics. Remodeling proteins and complexes are capable of sliding and evicting nucleosomes or exchanging histone subunits to permit the recruitment and action of DNA binding proteins and complexes on targeted regions (Saladi & de la Serna, 2010; Tang et al, 2010; Tyagi et al, 2016). Chromodomain-helicase-DNA-binding protein 7 (CHD7), homologous to protein KIS-L ("Kismet") in Drosophila, is an ATP-dependent chromatin remodeler which belongs to the CHD subfamily of trithorax-Group protein complexes (Srinivasan et al, 2008). CHD7 catalyzes the remodeling of nucleosomes in an ATP-dependent manner and plays a critical role in cell proliferation and differentiation (Layman et al, 2009; Bajpai et al, 2010; Bouazoune & Kingston, 2012). Mutations of the *Chd7* gene are considered to be drivers of CHARGE syndrome in humans, which causes several common developmental disorders including coloboma, heart defects, atresia choanae, growth retardation, genital abnormalities, and ear abnormalities (Vissers et al, 2004; Zentner et al, 2010). Numerous missense, nonsense and frameshift mutations identified from CHARGE patients distribute throughout the entire *Chd7* coding region which result in the haplodeficient CHD7 protein and misregulation of gene expression (Balasubramanian et al, 2014).

CHD7 is a large protein composed of 2,997 amino acids that contains annotated chromo-, SNF2-, helicase-, SANT-, and BRK-domains (Bouazoune & Kingston, 2012). The chromodomain of CHD7 binds preferentially to mono-methylated H3K4 in enhancer regions (Schnetz et al, 2009, 2010). As expected, because of having SNF2-helicase domains, CHD7 is able to remodel nucleosomes in vitro; however, it has characteristics distinct from several other remodeling classes, including the switch/sucrose non-fermentable (SWI/SNF) family of trxG remodelers (Bouazoune & Kingston, 2012). Although the functions of SANT- and BRK-domains are less clear,

---

[1]Department of Molecular Biology, Massachusetts General Hospital, Boston, MA, USA   [2]Department of Genetics, Harvard Medical School, Boston, MA, USA   [3]Department of Pathology, Massachusetts General Hospital and Harvard Medical School, Boston, MA, USA

Correspondence: kingston@molbio.mgh.harvard.edu

 

the potential interaction of CHD7 with polynucleotides and proteins through these domains has been proposed (Boyer et al, 2004; Allen et al, 2007, 2020; Ryan et al, 2011). Regulatory interactions and any additional functions for this protein remain relatively underexplored in part because of its large size which hinders analysis.

Long non-coding RNAs (lncRNAs) have been proposed to regulate many of the proteins and complexes involved in modulating chromatin structure. These lncRNAs are normally defined as being more than 200 nucleotides with no protein coding potential and are known to play important roles in organizing the structure of nucleolus, chromatin dynamics, and gene expression (Quinodoz & Guttman, 2014). For instance, the lncRNA XIST has been shown to target to one of X chromosomes and to recruit various proteins including members of the Polycomb-Group family to generate X chromosome inactivation (Loda & Heard, 2019). Several lncRNAs that are expressed in a lineage specific manner have been shown to regulate specific differentiation pathways in mammals (Hung & Chang, 2010; Sweta et al, 2019; Constanty & Shkumatava, 2021). LncRNAs contain information from the transcribed DNA template and therefore might use the specific recognition between sequences to guide or decoy the factors interacting with lncRNA to or away from the targeted genomic sites; the flexible structure of lncRNA might also enable the formation of various secondary conformations which could serve as a platform for interactions with proteins (Bhat et al, 2016). Although there is rarely a strong thermodynamic basis for specific RNA–protein interactions, there are several examples of contributions of sequence specificity to function (Davidovich et al, 2015; Statello et al, 2021). Because many lncRNAs are nuclear, spatial considerations and/or nuclear compartmentalization might also contribute to function and to specificity of interaction (Akkipeddi et al, 2020).

It has been estimated that about 43% of human genome is composed of repeat sequences and many are actively transcribed in a temporal and spatial specific manner (Goke & Ng, 2016). Repeat sequences of transposons derived from human endogenous retroviruses (HERV) comprise around 8% of the human genome and the subfamily-H (HERVH) is one of the most abundant members in this class (Yi & Kim, 2004; Santoni et al, 2012). Intact and full length HERVH family lncRNAs are composed of viral sequences for Pro, Gag, Pol and Env and have two LTRs flanking on each side and are expressed in pluripotent stem cells derived from inner cell mass but not in earlier embryonic stages or differentiated cells (Kelley & Rinn, 2012; Lu et al, 2014; Ohnuki et al, 2014; Wang et al, 2014a). Many members of the HERVH family include subsets of these sequences. The sequences of HERVH family members are evolutionarily conserved in primates (Ramsay et al, 2017). The mutations and deletions accumulated in HERVH copies over evolution have compromised the ability of HERVH to transpose in genome and to produce functional proteins for virus assembling (Mager & Freeman, 1995; Lindeskog et al, 1999; de Parseval et al, 2001). Previous studies have shown that HERVH family lncRNAs are important in regulating certain key pluripotency genes and maintaining cell pluripotency (Lu et al, 2014; Ohnuki et al, 2014; Wang et al, 2014a). HERVH lncRNAs have been shown to regulate long-range chromatin architecture, consistent with an interaction with chromatin modifying proteins (Zhang et al, 2019). However, the interactions that allow HERVH members to function in chromatin-based regulation of pluripotency and differentiation are not clear.

We screened for human RNAs that interact with CHD7 using PAR-CLIP and identified HERVH family lncRNAs as highly enriched interactors. We performed a detailed genomic mapping to examine CHD7 function in WT and HERVH-depleted ES cells and found increased CHD7 binding and H3K27ac levels on a subset of enhancers. Gene networks that were up-regulated by HERVH depletion corresponded to differentiation pathways. HERVH binds with high affinity but low specificity to CHD7 and inhibits remodeling activity, leading us to propose that HERVH plays a key role in modulating CHD7 activity in pluripotent cells via direct interaction.

# Results

## PAR-CLIP identifies RNAs interacting with CHD7 and HERVH RNA was enriched in CHD7 bound samples

To identify RNAs that interact with CHD7 in cells we used photo-activatable ribonucleoside-enhanced crosslinking and immuno-precipitation (PAR-CLIP [Spitzer et al, 2014]). PAR-CLIP was carried out with human embryonic stem cell (hESC) line H9 and an anti-serum specific for CHD7 (Fig S1A). Nascent RNA was labeled with 4-thiouridin (4-SU) to increase photo-crosslinking efficiency, immunoprecipitated with CHD7, and samples were separated using SDS–PAGE and transferred to a nitrocellulose membrane. The RNA was labeled with [γ-32P]-ATP labeling and an enriched signal representing CHD7–RNA complexes was observed that migrated above 250 KD (Fig 1A, left panel; note that the RNA–protein complexes formed are large and run indistinctly because of their size). RNAs were extracted from the CHD7–RNA complexes (Fig 1A, right panel) and made into libraries for deep sequencing. To identify peaks of PAR-CLIP enrichment, we first used the specialized PAR-alyzer method which takes into account the T to C nucleotide transitions that occur after direct RNA–protein interactions (Corcoran et al, 2011). Because many of the PARalyzer peaks had small magnitude, we validated the significance of PAR-CLIP signal enrichment over input with the widely used SPP peak caller (Kharchenko et al, 2008) and a cutoff of absolute read density within a peak (reads per kilobase per million [RPKM] > 10). PARalyzer called "read clusters" as peaks, many of which were multiple narrower peaks within larger peaks called by SPP (example shown in Fig S1B). All 1,475 resulting SPP peaks that were consistently called in two biological replicates (Fig 1B) overlapped with PARalyzer peaks and had a substantial ~10% rate of T to C conversions among reads (Fig S1C). Therefore, we used these 1,475 peaks as a stringent set that satisfied the criteria of both the presence of T to C conversions and strong enrichment of PAR-CLIP signal over input. These peaks are distributed throughout the genome with the majority mapping to coding regions and substantive numbers also mapping to annotated long intergenic non-coding RNAs (lincRNAs) and intergenic regions (Fig S1D). These enriched peaks were mapped to 862 annotated genes over 77% of which were actively transcribed (RNA-seq reads per kilobase per million [RPKM] > 1) (Fig S1E). Although transcripts with higher expression level often showed stronger enrichment than lower expressed transcripts, only about 10% of highly expressed transcripts (RPKM > 10–50) were

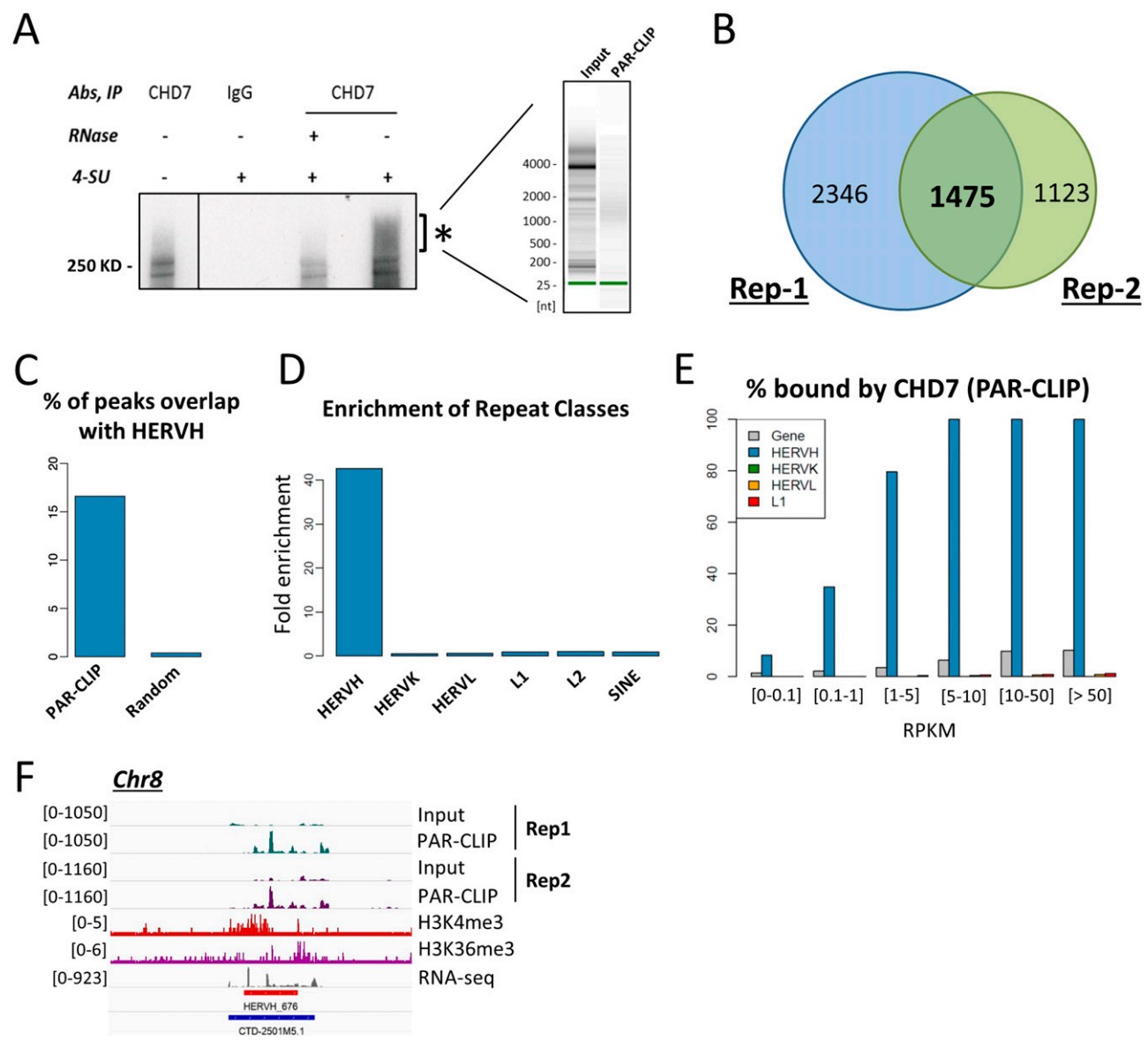

**Figure 1. PAR-CLIP identifies the HERVH transcript as a major RNA interacting with CHD7 in hESCs.**
**(A)** RNA-protein complexes with decreased mobility were enriched by CHD7 antibody pull-down (left panel, indicated by asterisk). Labeled RNA was extracted and purified from the identified RNA-protein complexes (right panel) and sequenced. **(B)** 1,475 RNA transcripts which showed strong enrichment over input from antibody pull-down were identified in two PAR-CLIP replicates. **(C)** RNA transcripts identified from PAR-CLIP were analyzed by RepeatMasker. The same number of random genomic loci was analyzed for comparison. **(D)** HERVH was significantly enriched by PAR-CLIP in contrast to other types of repeat sequences in the human genome, including HEVHL, LINE (L1 and L2), and SINE. **(E)** Enrichment of RNA in PAR-CLIP was positively correlated with expression level; enrichment of HERVH was more pronounced than the coding genes and other repeat elements expressed at a similar RPKM level. **(F)** A representative example of enriched HERVH RNA by PAR-CLIP; the repeat element shown is on chromosome 8 and overlaps with an annotated long non-coding RNA CTD-2501M5.1.

enriched in CHD7 PAR-CLIP signal compared to the input background (Fig S1F). We analyzed some highly expressed genes, for example, SRSF3 and HNRNPA1, and we did not observe significant enrichment of these RNA transcripts over input (Fig S1G). We conclude based upon these data that RNAs identified using PAR-CLIP are candidates for interaction with CHD7 and are not simply reflective of their expression level.

To analyze whether any repeat elements are enhanced in CHD7-bound RNAs, we cross-referenced the PAR-CLIP data with genomic repeat annotations as determined by RepeatMasker (Smit et al,

2013-2015). Interestingly, about 17% of the enriched peaks were transcribed from the human endogenous retrovirus subfamily-H (HERVH) loci (Fig 1C). When we considered all 175,928 PARalyzer peaks, without additional filtering by the magnitude of PAR-CLIP signal using SPP, this less stringent larger peak set was also strongly enriched for the presence of HERVH elements (Fig S1H). No other classes of repeat sequences, for example, LINE, SINE, or other HERV types such as HERVK and HERVL were enriched (Fig 1D). In hESCs, HERVH repeats have relatively higher expression (>1 RPKM on average for ~1,200 copies) than other types of repeats (~0.2–0.3 RPKM

on average for all copies) (Fig S1I). To assess whether the observed enrichment of PAR-CLIP signal among HERVH transcripts could be attributed to this stronger expression, we analyzed the correlation of PAR-CLIP signal with varying expression levels among individual HERVH copies. Most of the HERVH copies with a robust expression level of RPKM >1 showed enrichment after CHD7 PAR-CLIP (Fig 1E and Table S1). Notably, close to 40% of the lower expressed alleles of HERVH (RPKM of 0.1–1) were also enriched (Table S1). This incidence of CHD7-bound transcripts among HERVH family members was significantly higher than among coding genes, including coding genes with much higher expression (Fig 1E). Thus, regardless of the expression level, HERVH transcripts preferably interact with CHD7 in hESCs. In addition, most of the HERVH genomic loci that were enriched in the CHD7 PAR-CLIP are decorated with histone marks H3K4me3 and H3K36me3, consistent with their active state of transcription and as seen by others previously (Fig 1F, exemplified with one HERVH element that, as with many HERVH elements is found in a position that overlaps with an annotated lncRNA, in this case one called CTD-2501M5.1) (Kelley & Rinn, 2012; Santoni et al, 2012; Wang et al, 2014a). These analyses established HERVH as a candidate for functional interactions with CHD7 that might impact regulation in human ES cells.

## CHD7 binds to enhancers and correlates with H3K27ac modification

To examine whether HERVH impacts CHD7 function, it was necessary to first characterize CHD7 localization in hESCs. The function of chromatin remodelers, as with most regulatory factors, is determined in part by targeting to specific chromatin regions. We used CUT&RUN to characterize the binding of CHD7 to chromatin (Skene & Henikoff, 2017). To improve the specificity of antibody recognition of CHD7, we incorporated an HA-tag on the C terminus of the endogenous CHD7 gene using CRISPR-Cas9. The expression of HA-tagged CHD7 protein in hESCs was found to be similar to untagged CHD7 as measured by Western blotting and its nuclear localization was confirmed by immunostaining with HA and CHD7-specific antibodies, respectively (Figs 2A and S2A). As a negative control in localization experiments, we generated CHD7 knockout cells by introducing either InDel mutations or a stop codon in exon-2 of Chd7 (Fig S2B). Both mutations displayed similar and overlapping effects on gene expression (Fig S2C); we used the stop codon mutation in the experiments presented below. The absence of CHD7 in cells was verified by Western blotting and immunostaining (Figs 2A and S2A and D). The expression levels of key pluripotent and differentiated genes were measured using RNA-seq to confirm that the CHD7 knockout cells remained in a pluripotent state (Fig S2E). We performed CHD7 CUT&RUN with both HA and CHD7-specific antibodies. Both antibodies detected CHD7 binding regions similarly (Fig 2B, upper panel) and peaks obtained from both datasets showed significant overlap (Fig 2B, lower panel). The CHD7-specific antibody normally produced data with better signal-to-noise ratio (Fig S3A) and hence more peaks were identified than in HA antibody. Importantly, both antibodies showed high specificity to CHD7 and we observed very few peaks in CHD7 knockout cells as compared with WT (Fig S3B). These results validate these reagents and this methodology in identifying CHD7 bound regions in the genome.

We analyzed the genomic distribution of CHD7 using data obtained with the CHD7-specific antibody. Among 6,502 CHD7 peaks, the majority (~80%) overlapped with ENCODE enhancers annotated using ChromHMM based on chromatin marks (Fig 2C) (Ernst et al, 2011; ENCODE Project Consortium, 2012). We performed CUT&RUN for H3K27ac, the histone modification of active enhancers, and found that it was strongly enriched among the enhancers that overlapped CHD7 peaks (Fig 2D, P-value < 1 × 10$^{-100}$). Similar to peaks identified by CHD7 antibody, peaks identified by HA antibody and peaks of H3K27ac largely overlapped with ENCODE enhancers (Fig S3C). Furthermore, CHD7 was enriched among hESC enhancers, and 36.4% (38,778/106,616) of them were bound by CHD7 with at least 1.5-fold enrichment over genomic background (Fig 2E). The intensity of CHD7 binding was positively correlated with the intensity of H3K27ac on enhancers which is in agreement with the previously reported association of CHD7 with active enhancers (Fig 2E) (Schnetz et al, 2010).

To examine whether CHD7 occupancy had an impact on H3K27ac decoration at CHD7 bound enhancers, we analyzed the effect of CHD7 depletion on the level of H3K27ac at CHD7 target regions. We examined all ENCODE enhancers and identified 2,880 enhancers where the H3K27ac level was consistently altered by CHD7 knockout in four replicates (Figs 2F and S3D). Notably, H3K27ac levels were altered in both directions by CHD7 knockout, with 2,209 enhancers showing increased H3K27ac ("Up" enhancers) and 671 enhancers showing decreased H3K27ac ("Down" enhancers) following CHD7 knockout. In both cases, enhancers whose H3K27ac levels were altered overlapped with those bound by CHD7 (Fig 2G). These data showed an impact of CHD7 depletion on H3K27ac levels but the direction of the change differed for distinct classes of enhancers.

We examined various characteristics of the "Up" and "Down" sets of enhancers to attempt to understand their differences in response to CHD7 knockout. In particular, we analyzed the occupancies of CHD7 and p300, the protein primarily responsible for acetylation of H3K27, as well as the densities of histone modifications at these two sets of enhancers. The level of CHD7 binding was in general higher at "Down" enhancers where H3K27ac decreased upon CHD7 knockout, when compared with "Up" enhancers where H3K27ac increased (Fig 2H). This result was consistent in all four replicates of experiments performed (Fig S3E). This is consistent with CHD7 contributing to H3K27 acetylation on a subset of enhancers where it binds at high levels. An example of genomic tracks at one of these enhancers is shown in the left panel of Fig 2I; a similar effect among top 100 CHD7 bound enhancers is shown in Fig S3F. The right panel of Fig 2I also shows an example of an enhancer with lower level of CHD7 that displayed increased H3K27ac in the absence of CHD7. The effect of CHD7 knockout on H3K4me1, another histone modification often presented at enhancers, was similar to the effect on H3K27ac (Fig S3G). The total level of H3K27ac and H3K4me1 was not significantly affected by CHD7 depletion as compared with WT (Fig S3H). We determined whether p300 was impacted by CHD7 depletion. Consistent with its biochemical role, binding of p300 decreased on enhancers with reduced H3K27ac and increased on enhancers with increasing H3K27ac after CHD7 knockout (Figs S3I and 2I). Surprisingly, the total level of p300 was elevated in CHD7 knockout cells (Fig S3J), which might be one reason for the increased acetylation on a subset of

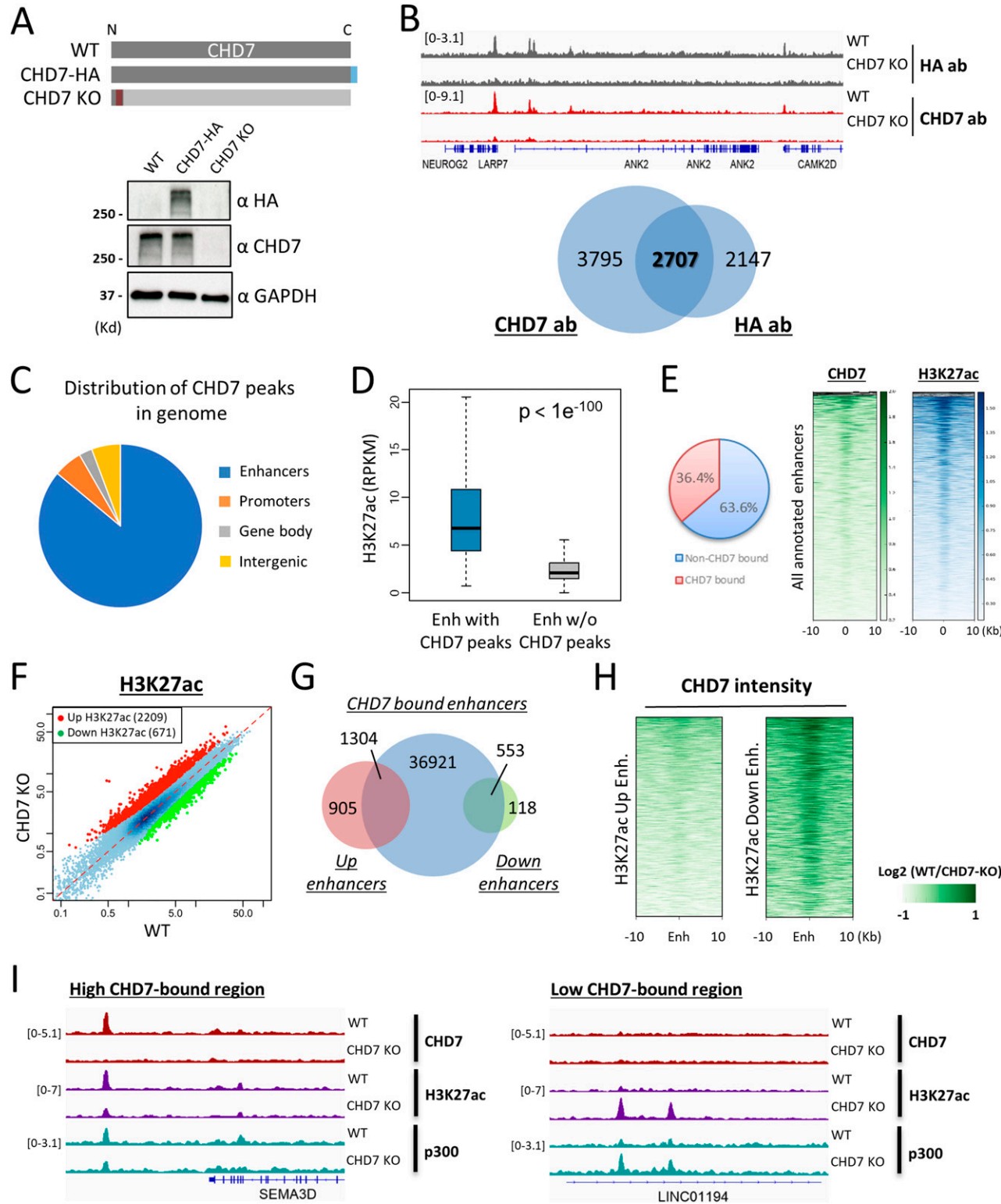

**Figure 2.   CHD7 preferentially targets enhancer regions and knockout alters levels of H3K27ac.**
**(A)** CRISPR-Cas9 was used to insert a HA-tag at the C-terminus of CHD7 protein and to create a CHD7 null mutant in hESCs. Expression levels of CHD7 protein were measured by CHD7-specific and HA antibodies. **(B)** Upper panel: a representative example of CUT&RUN-seq done by HA and CHD7 antibodies in WT and CHD7 knockout cells; lower panel: the overlap of peaks identified from HA and CHD7 antibodies. **(C)** Analysis of CHD7 bound peaks reveals that the majority of them (~80%) overlap with enhancers annotated by ENCODE. **(D)** CHD7 bound enhancers are significantly enriched with acetylation on H3 Lysine-27 comparing with random enhancers (P-value < 1 × 10$^{-100}$, by two-tailed t test). **(E)** 36.4% of annotated enhancers (38,778/106,616) were bound by CHD7 with at least 1.5-fold enrichment over genomic background and the

enhancers in those cells. P300 has been shown to interact with CHD7, which is consistent with the potential for CHD7 to assist with p300 binding on the enhancers where H3K27ac decreased with CHD7 knockout. These previous studies also observed p300 functioning both cooperatively with and independently from CHD7, as we saw in our analysis (Schnetz et al, 2010).

To evaluate the association between the effects of CHD7 on H3K27ac levels at enhancers and expression from proximal promoters, we analyzed the correlation between enhancers with altered H3K27ac and the nearby differentially expressed genes (DEGs) in CHD7 knockout cells. Enhancers with up- and down-regulated levels of H3K27ac in the CHD7 depletion background showed strong enrichment of their positioning within a 100 kb vicinity of up- and down-regulated DEGs, 6.7-fold and 4.7-fold, respectively, in comparison to the same number of random enhancers (Fig S3K). Functional pathway enrichment analysis using EnrichR (Chen et al, 2013; Kuleshov et al, 2016) suggested that genes down-regulated in CHD7 knockout cells were enriched for regulatory networks, e.g., MTA1 and TWIST1, related to CHD7 according to the TRRUST database (Fig S3L) (Bajpai et al, 2010; Engelen et al, 2011; Han et al, 2018).

In summary, CHD7 plays a role in regulating H3K27 acetylation on a subset of enhancers where it binds strongly, perhaps via interplay with p300. CHD7 shows a more complex impact on enhancers where it is bound at lower levels, potentially through indirect effects which might include effects due to changes in overall p300 levels.

## HERVH RNA negatively regulates CHD7 targeting to enhancers

Having established the relation between CHD7 and H3K27ac, we investigated whether HERVH RNA impacted CHD7 binding and H3K27ac levels by using locked nucleic acid (LNA)-GapmeR antisense oligonucleotides (Pendergraff et al, 2017) to knockdown the expression of HERVH in hESCs and analyzing CHD7 and H3K27ac binding patterns using CUT&RUN. Three specific LNA-GapmeRs were designed that target the consensus sequences of HERVH elements in the Gag and Pro regions (Fig S4A, indicated by asterisks) and LNA-GapmeRs with no target in the human genome or targeting MALAT1 were used as controls. HERVH-specific LNA-GapmeRs specifically reduced HERVH expression by ~30% after 24 h of treatment, whereas the MALAT1-specific LNA reduced MALAT1 expression by ~70% and HERVH expression levels remained similar to control levels (Fig S4B). Longer time periods of LNA treatment did not substantively increase the extent of knockdown and resulted in increased levels of differentiation, a known effect of HERVH depletion (Lu et al, 2014; Wang et al, 2014a). We therefore chose to study the 24-h time point to increase the likelihood that effects observed might be direct. The control LNA did not affect either HERVH or Malat1 transcripts (Fig S4B). These data indicated that HERVH down-regulation by LNAs is specific.

To confirm the knockdown of individual HERVH copies and analyze its genome-wide effect on gene expression, we performed strand-specific RNA-seq in HERVH knockdown cells. HERVH repeats show sufficient variability in sequence that they can be distinguished from one another by mapping RNA-seq reads. Of the ~1,200 copies of annotated HERVH in the human genome (Wang et al, 2014a), ~10% are moderately or highly expressed with RPKM > 1, of which 51 were significantly reduced by HERVH-specific LNA-knockdown (Fig 3A, red dots representing logFC > 1, Table S1). We examined the expression levels of pluripotent genes, for example, *POU5F1*, *SOX2*, and *Nanog* and found that they were similar between WT and HERVH knockdown cells, whereas differentiated marker genes in all three developmental germ layers were not detected (Fig S4C). We conclude that transient treatment by LNAs to reduce HERVH RNA levels in hESCs did not substantively alter their pluripotent state.

We performed CUT&RUN for H3K27ac with HERVH knockdown cells. By comparing with WT, we identified 323 and 268 enhancers with up- and down-regulated H3K27ac, respectively, in HERVH knockdown cells (Fig 3B). We examined the CHD7 level of these enhancers and found that enhancers with up-regulated H3K27ac collectively showed a modest but appreciable increased CHD7 binding upon HERVH knockdown (P-value = $7.71 \times 10^{-3}$). In contrast, enhancers with decreased H3K27ac had similar levels of CHD7 as compared with before knockdown (Fig 3C). The increased binding of CHD7 to enhancers with up-regulated H3K27ac in HERVH knockdown was consistent in three replicates (Fig 3D, metaplot on the left). We examined individual enhancers with increased H3K27ac after HERVH knockdown and compared their CHD7 occupancy between WT and HERVH knockdown cells (Fig 3D, heatmap in the middle, logFC(HERVH KD/WT)). About 60% of examined enhancers displayed increased CHD7 binding (Fig 3D, highlighted by red-dotted rectangle, P-value = 0.012). These data indicated that HERVH has a negative effect on CHD7 targeting which directly or indirectly affects H3K27ac levels at a subset of enhancers.

To determine whether the observed up-regulation of H3K27ac induced by HERVH knockdown was CHD7-dependent, we used CHD7 knockout cells and found that H3K27ac levels at the same set of enhancers was no longer increased upon HERVH knockdown, indicating that CHD7 knockout was epistatic to HERVH knockdown at these enhancers (Fig 3D, heat map to the right, logFC(HERVH KD; CHD7 KO/CHD7 KO)). We wondered whether the intensified binding was reflected in a change in CHD7 protein level. We examined CHD7 protein by immunoblotting with CHD7-specific and HA antibodies and found no noticeable difference between WT and HERVH knockdown cells (Fig 3E, quantification was normalized to β-Actin expression). Thus, the physical interaction between CHD7 and HERVH RNA might have a direct impact on CHD7 targeting to chromatin. We examined the total level of H3K27ac and H3K4me1 and both of them remained at similar level in WT and HERVH knockdown cells (Fig 3E, quantification was normalized to H3). There was no impact on H3K4me1 levels on enhancers when WT and HERVH knockdown cells were compared (Fig S4D). Chromatin

intensity of CHD7 binding was positively correlated with the intensity of H3K27ac on enhancers. **(F)** The levels of H3K27ac of 2,209 and 671 enhancers were up- or down-regulated, respectively, after CHD7 knockout. **(G)** Enhancers with altered H3K27ac level in CHD7 knockout cells overlap with CHD7 bound enhancers identified in WT cells, especially enhancers of down-regulated H3K27ac. **(H)** Enhancers with down-regulated H3K27ac in CHD7 knockout cells had high CHD7 binding in WT cells, whereas enhancers with up-regulated H3K27ac had low CHD7 binding in WT cells. **(I)** Examples of enhancers where H3K27ac and p300 levels were dependent on (panel on the left) or independent from (panel on the right) CHD7 occupancy.

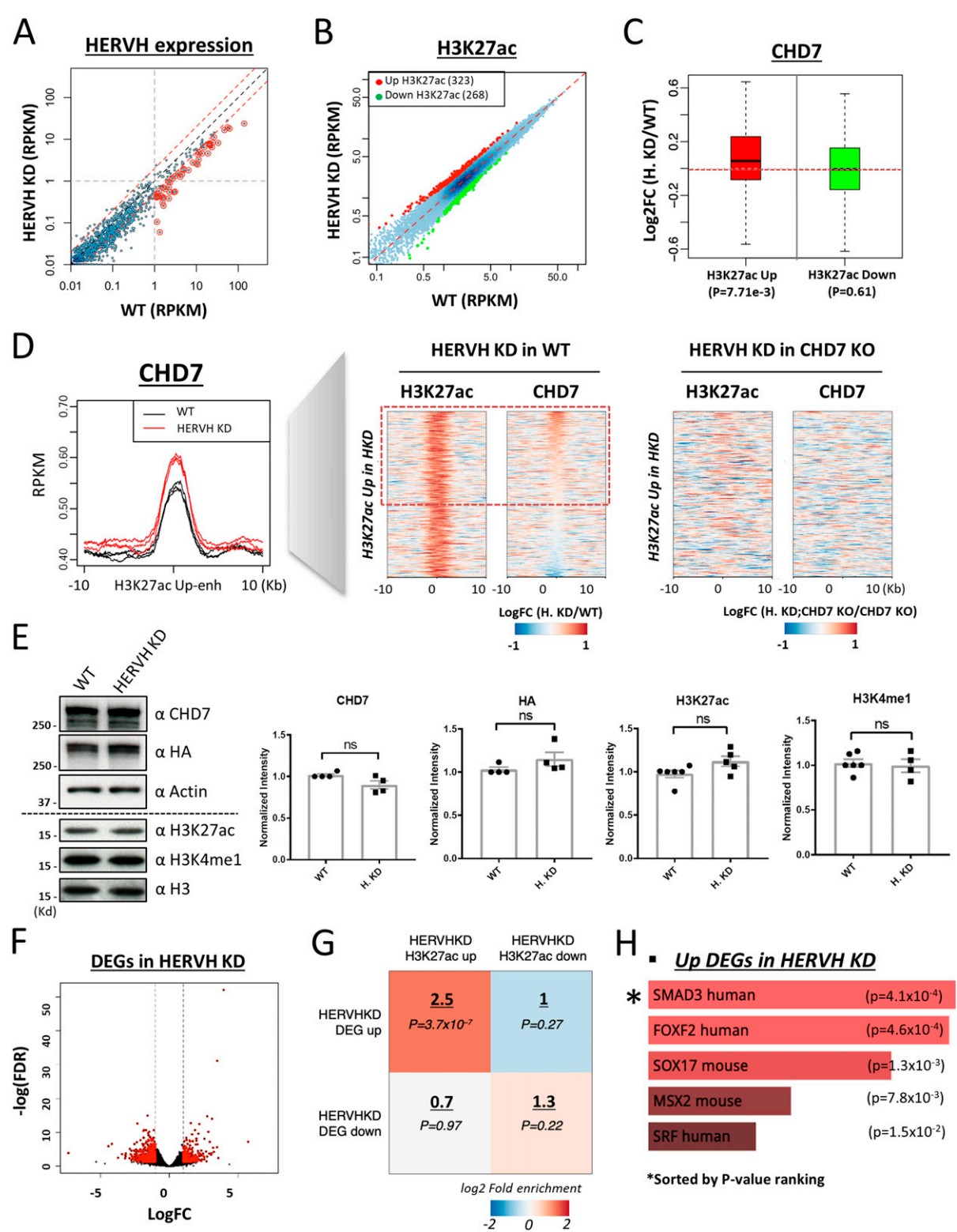

**Figure 3. Knockdown of HERVH expression by locked nucleic acid-GapmeRs affects CHD7 binding on enhancers.**
**(A)** The expression of all HERVH repeat elements was examined using RNA-seq to compare HERVH knockdown cells with WT. Most of the well-expressed HERVHs at RPKM > 1 was reduced more than twofold upon locked nucleic acid knockdown (labeled in red). **(B)** The levels of H3K27ac at 323 and 268 annotated enhancers were up- and down-regulated, respectively, in HERVH knockdown cells. **(C)** CHD7 binding levels changed on enhancers with H3K27ac levels that were altered upon HERVH knockdown. *P*-values were determined by two-tailed *t* test. **(D)** Left panel: increased CHD7 binding to enhancers with increased H3K27ac when HERVH expression was knocked down; Middle panel: H3K27ac and CHD7 binding levels for individual enhancers comparing WT and HERVH knockdown cells; Right panel: lack of change in H3K27ac and CHD7

accessibility was measured and we observed no significant difference between WT and HERVH knockdown (Fig S4E).

Taken together, these observations suggest a model in which the interaction of CHD7 and HERVH RNAs inhibits its binding to target sites and subsequently impacts acetylation of H3K27 at a subset of enhancers.

## DEGs induced by HERVH knockdown were enriched near H3K27ac altered enhancers

Altering HERVH expression influenced CHD7 binding and epigenetic dynamics on enhancers genome wide. We wondered how this alteration would impact gene expression. We analyzed transcription in HERVH knockdown cells by RNA-seq and identified 415 up- and 566 down-regulated DEGs, respectively, which had >2-fold change in expression compared with WT (Fig 3F). We performed enrichment analysis to examine the proximity of these DEGs to enhancers affected by HERVH knockdown. We found that up-regulated H3K27ac enhancers were enriched within 100 kb of up-regulated genes (2.5-fold enrichment comparing to same number of random enhancers, $P$-value = $3.7 \times 10^{-7}$) and minimal correlations were observed for enhancers with down-regulated H3K27ac activity and down-regulated DEGs (Fig 3G). We conclude that the primary regulatory impact of HERVH depletion in these cells is to up-regulate a set of genes near enhancers where WT HERVH expression leads to decreased acetylation and decreased CHD7 binding (Fig 3C and D). We analyzed the pathways that were impacted on this set of genes. Analysis of enrichment of functional gene categories using EnrichR suggested that the up-regulated genes were enriched in several pathways including FOXF2 and the SMAD3-controlled genes listed in the TRRUST database (Fig 3H). Both FOXF2 and SMAD pathway genes are involved in differentiation processes (Zhang et al, 1998; Shen, 2007; Liu et al, 2014; Wang et al, 2014b). This raises the possibility that HERVH might tune expression of genes via its interaction with CHD7 that are involved in differentiation; however, that hypothesis would require further analysis across much lengthier time courses of differentiation than achieved in this analysis, which examines only changes within the first 24 h of depletion.

## Deletion of an individual HERVH led to a minor but global effect of CHD7 targeting

Certain HERVH sequences are highly conserved. For instance, analysis of *env* fragments of HERVH showed that they share 82–99% sequence similarity among each other (Yi & Kim, 2004). It is possible that the expressed HERVH repeats all contribute to a general trans effect or, alternatively, that the large number of loci expressing HERVH leads to a widespread effect due to a large number of cis acting sequences. Previous work has shown that deletion or silencing of individual actively transcribed HERVH impacts the establishment of topologically associating domains and consequently affects the transcription profile of nearby regions (Zhang et al, 2019). To further examine how HERVH affects CHD7 targeting, and to understand whether HERVH RNAs function solely in cis or whether disruption of one of the repeats has broader impact, we deleted an individual HERVH gene using CRISPR-Cas9.

We deleted an HERVH element which is well expressed (RPKM 7.0 by RNA-seq) from an intergenic location on chromosome 2 (Chr2): 210,164,037–210,169,656. The RNA transcribed from this HERVH element was found to interact with CHD7 as assessed by PAR-CLIP analysis (Fig 4A). Two CRISPR-Cas9 guide-RNAs were designed that target unique sequences outside of the HERVH genomic locus, thereby resulting in deletion of the entire HERVH region (Fig 4B). Homozygous knockout clones were screened and confirmed by PCR with specific external and internal primers and two independent HERVH knockout clones were obtained (Figs 4B and S5A). RNA-seq showed depletion of expression from the deleted HERVH repeat in knockout cells (Fig 4C, deleted HERVH element circled in red). Deletion of this specific HERVH locus did not affect the expression of most of others HERVH repeats, although it did result in depletion of two HERVH repeats located on chromosomes 4 and chromosome 16 (Fig 4C). The total expression level of HERVH measured by qPCR in HERVH element deleted cells was mildly reduced, as expected for knockout of a single copy of the repeats (Fig S5B). There was no apparent morphological difference between WT and HERVH element deleted cells. The estimated expression of key pluripotent genes remained at a similar level as in WT and the expression of differentiation marker genes was similar and at a low or undetectable level in both WT and HERVH knockout cells (Fig S5C). To examine whether there was evidence for changes in these cells along differentiation pathways, we used quantitative-PCR to assess expression of the neuronal differentiation gene PAX6. We detected a slight up-regulation of PAX6 in HERVH element deleted cells (Fig S5D); however, the expression of PAX6 in these HERVH knockout cells is over 100-fold less than in neural progenitor cells differentiated from hESCs.

We determined whether deleting a single HERVH element would have effects similar to those observed when HERVH expression was knocked down with LNAs. We note that the time frame of these two experiments differ; effects of LNA knockdown were purposefully measured at 24 h with the aim of examining direct effects, whereas the selection of clonal knockout lines takes ~1 mo, leading to possible impacts of secondary effects that might occur over this longer time frame. We measured H3K27ac levels by CUT&RUN and there were 234 and 360 enhancers with increased or decreased H3K27ac intensities, respectively, when those levels in a single HERVH KO were compared with WT (Fig 4D). We measured CHD7 occupancy on these enhancers and found that enhancers with increased H3K27ac had increased CHD7 binding and that enhancers

binding levels for individual enhancers between CHD7 knockout cells when HERVH is depleted. **(E)** CHD7 protein and histone modifications in HERVH knockdown cells. CHD7 protein level was detected by CHD7 and HA antibodies and normalized to Actin and the levels of H3K27ac and H3K4me1 were normalized to H3 for quantification. *P*-value thresholds were assessed using unpaired *t* test. Ns, nonsignificant, *P* > 0.05. **(F)** Differentially expressed genes (DEGs), either up- or down-regulated (indicated by red dots), in HERVH knockdown cells in contrast to WT (logFC > 1). **(G)** Enrichment of enhancers with altered H3K27ac intensity and DEGs within 100 kb of the enhancers in HERVH knockdown cells. The enrichment of fold-change underscored was determined by comparing proximal differential H3K27ac enhancers to proximal random enhancers. *P*-values were determined by two-tailed *t* test. **(H)** Functional analysis by EnrichR identified candidate pathways for the up-regulated DEGs in HERVH knockdown cells.

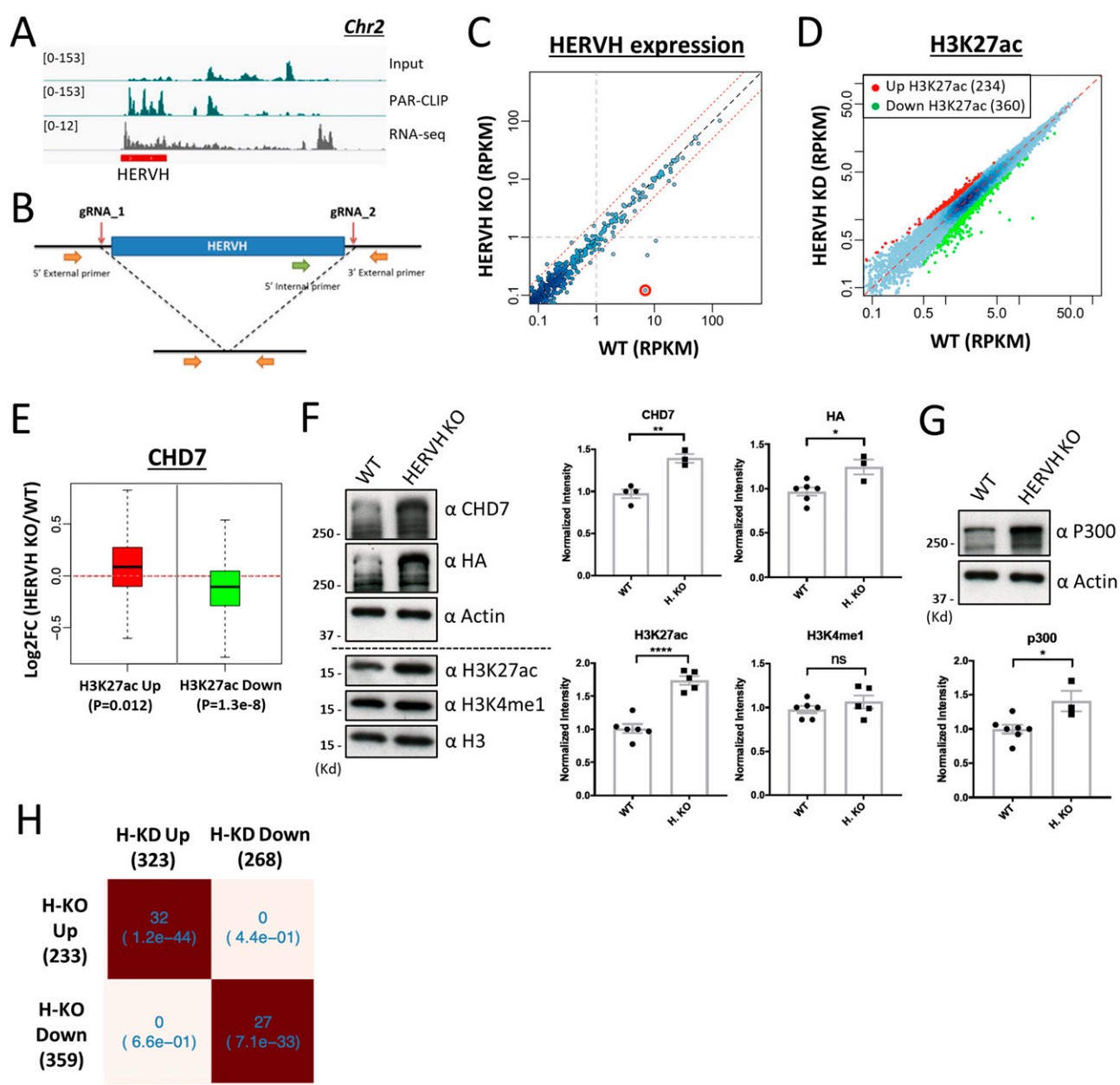

**Figure 4. Individual HERVH deletion leads to widespread changes of H3K27ac intensity and CHD7 occupancy on enhancers.**
**(A)** The HERVH element on chromosome-2 which was well-expressed and was enriched by PAR-CLIP to CHD7. **(B)** Deletion of the HERVH element by CRISPR-Cas9 with two guide RNAs recognizing unique sequences outside of the repeat element. Internal and external primer sets were used to screen for the HERVH deleted cells. **(C)** Expression of all HERVH repeat elements was examined by RNA-seq in HERVH knockout cells compared with WT. Deleted HERVH labeled in red. **(D)** Levels of H3K27ac of 234 and 360 enhancers were up- and down-regulated, respectively, in HERVH deleted cells. **(E)** CHD7 binding on enhancers with altered H3K27ac levels in the HERVH knockout background. **(F)** CHD7 protein and histone modifications in HERVH knockout cells. The CHD7 protein level, detected by CHD7 and HA antibodies, was normalized to Actin and the levels of H3K27ac and H3K4me1 were normalized to H3 for quantification. *P*-value thresholds were assessed using unpaired *t* test. *$P < 0.05$; **$P < 0.01$; ****$P < 0.0001$; ns: nonsignificant, $P > 0.05$. **(G)** P300 protein level in HERVH knockout cells; Western blots were normalized to Actin for quantification. *$P < 0.05$. **(H)** Enhancers affected by HERVH knockdown and HERVH knockout were positively correlated. *P*-values were determined by two-tailed *t* test.

with decreased H3K27ac had decreased CHD7 binding in HERVH element–deleted cells compared with WT (Figs 4E and S5E). These data are consistent with the results reported above showing that the changes of H3K27ac level triggered by altering HERVH expression were positively correlated with the binding of CHD7 on targeted enhancers.

To explore further the relationships between CHD7 and H3K27ac in these HERVH deletion cells, we measured levels of overall CHD7, H3K27ac, and p300. CHD7 protein level was measured by immunoblotting with CHD7-specific and HA antibody in HERVH element deleted cells and was found to be elevated as compared with WT (Fig 4F, upper panel, quantification was normalized to β-Actin).

Thus, the global increase in binding of CHD7 in these cells might be caused, at least in part, by increased CHD7 levels. Measurement of the total level of H3K27ac and H3K4me1 revealed that H3K27ac level was significantly increased in HERVH element deleted cells, whereas H3K4me1 level remained similar to WT (Fig 4F, lower panel, quantification was normalized to H3). To determine whether the overall increase in H3K27ac correlated with an increase in total p300, we measured p300 levels and found that they were also up-regulated in the HERVH element–deleted background (Fig 4G, quantification was normalized to H3). These results offer further data supporting the presence of regulatory loops in which HERVH modulates CHD7, p300, and H3K27ac. H3K4me1 occupancy on the chromosome was analyzed and minimal changes were observed on H3K27ac up- and down-regulated enhancers in HERVH element deleted cells (Fig S5F). ATAC-seq analysis of enhancers with altered H3K27ac intensity showed mild but corresponding changes of increasing or decreasing of their chromatin accessibility (Fig S5G).

To investigate whether the impact of deleting a single HERVH copy occurred in cis or in trans, we analyzed whether the effects had a strong bias to Chr2 (where the deletion was made) or impacted other chromosomes. We evaluated changes of CHD7 binding for individual chromosomes. In HERVH knockout cells, CHD7 levels were increased at 29 enhancers and reduced at 70 enhancers. The genomic distribution of the affected enhancers did not show any particular clustering and was not linked to the specific location of the ablated HERVH copy or even its chromosome (Chr 2, which had only five enhancers with increased CHD7 binding and 10 enhancers with reduced CHD7 binding). Similarly, the genomic distribution of enhancers with affected H3K27ac levels (234 up-regulated and 360 down-regulated enhancers genome-wide) did not have a strong bias towards either the location of the ablated HERVH or chromosome 2 in general (13 up-regulated and 14 down-regulated enhancers on Chr2). These data suggested that depletion of a specific HERVH element expression can influence CHD7 binding not only on chromosome where this HERVH element was resided in but also on other chromosomes.

We analyzed the transcriptome of HERVH element deleted cells. Twenty-three down-regulated DEGs were identified compared with WT and five of them were overlapped with DEGs of HERVH knockdown (Fig S5H, $\log_2$FC > 1, red open circle). We did not find any significantly enriched functional gene categories within this small number of DEGs in HERVH removal cells. We found that 14% (32/233) and 7.5% (27/359) enhancers with up- or down-regulated H3K27ac in HERVH element–deleted cells overlapped with enhancers having up- or down-regulated H3K27ac in HERVH knockdown cells, respectively (Fig 4H). These data are consistent with the hypothesis that deletion of a single HERVH impacts a subset of enhancers and genes as compared with knockdown of all HERVH copies. The largely unchanged transcriptome reaffirmed that HERVH element deleted cells still remained in a state similar to WT, even though the levels of CHD7 and H3K27ac were elevated (Figs 4F and S5B).

In summary, we found that deletion of an individual HERVH led to broad changes in CHD7 targeting and H3K27ac occupancy. This deletion also caused changes in the overall amount of CHD7, p300, and H3K27ac in these cells, indicating that the interaction between HERVH and CHD7 might impact function at individual genomic sites and also impact cross-regulation of the amount of protein and histone modifications in this network.

## HERVH RNA has high binding affinity but low specificity to CHD7

The PAR-CLIP experiments showed that HERVH is preferentially bound by CHD7 in cells. This specific interaction identified in cells might be based on a sequence-dependent mechanism, it might require the involvement of other proteins to generate the binding specificity, or it might result from spatial proximity in the nucleus. To address whether there is any sequence specificity in binding between CHD7 and HERVH, we used electrophoresis mobility shift assays. To generate increased resolution, we divided an ~6.5-kb full-length HERVH element into eight smaller segments, each ~0.8 kb long (Fig 5A, upper panel). We chose a specific HERVH element located on chromosome 8 which overlapped with an annotated lncRNA CTD-2501M5.1 and which was well represented in the PAR-CLIP analysis (Fig S6A). This HERVH element was isolated by cloning following PCR amplification and individual segments were subcloned adjacent to a T7 promoter for expression of Cyanine 5 (Cy5)-UTP labeled RNA subfragments (Fig 5A, lower panel).

Binding analysis was performed by incubating 1 nM of labeled RNA with increasing amounts of purified full-length CHD7 protein and the bound complexes were separated in the native agarose gel for analysis. In all tested segments, HERVH RNA bound to CHD7 as shown by specifically shifted species in gel (Fig S6B). We quantified the gels and calculated the binding affinity (Kd, 50% binding saturation) of each of RNA segments to protein. All HERVH RNA segments had similar binding affinity to CHD7 (Fig 5B). Among those, the S8 segment presented the strongest binding (Kd 7.7 ± 0.19 nM) and the S5 segment presented the weakest binding (Kd 13.1 ± 0.56 nM) to CHD7, small differences that seem unlikely to be biologically significant. To serve as a control, GFP RNA which has about the same length to HERVH RNA segments was tested. The binding affinity of GFP RNA to CHD7 protein was similar to HERVH RNA segments (Fig S6C). We examined whether the interaction between HERVH and CHD7 affected binding of CHD7 to a nucleosome. We reconstituted a Cy-5 labeled mononucleosome, incubated with excess CHD7, and observed a complete shift of the mononucleosome to two slower migrating species (Fig 5C, lane 3). We titrated in varying amounts of HERVH RNA and found that CHD7 was competed away and the mononucleosome band was observed (Fig 5C). We also observed a faster migrating band at the size expected for a hexameric nucleosome. It is possible that either CHD7 binding or the excess of HERVH RNA might contribute to removal of an H2A-H2B dimer from the normal octameric nucleosome.

Taken together, the data indicated that HERVH RNA has high binding affinity but low specificity to CHD7 in the in vitro binding setting and additional factors and/or subnuclear localization might be necessary to achieve a specific interaction between HERVH RNA and CHD7 as was observed in cells. This result obtained from the competition assay is line with our studies in cells that the presence of HERVH RNA decreases binding of CHD7 to chromatin.

## Effect of HERVH knockdown on CHD7 binding near HERVH alleles

Given the lack of specificity of HERVH binding to CHD7 observed above, we used our in vivo analysis of CHD7 binding patterns to test the hypothesis that binding of HERVH to CHD7 is increased by proximity. We took advantage of the observation that highly

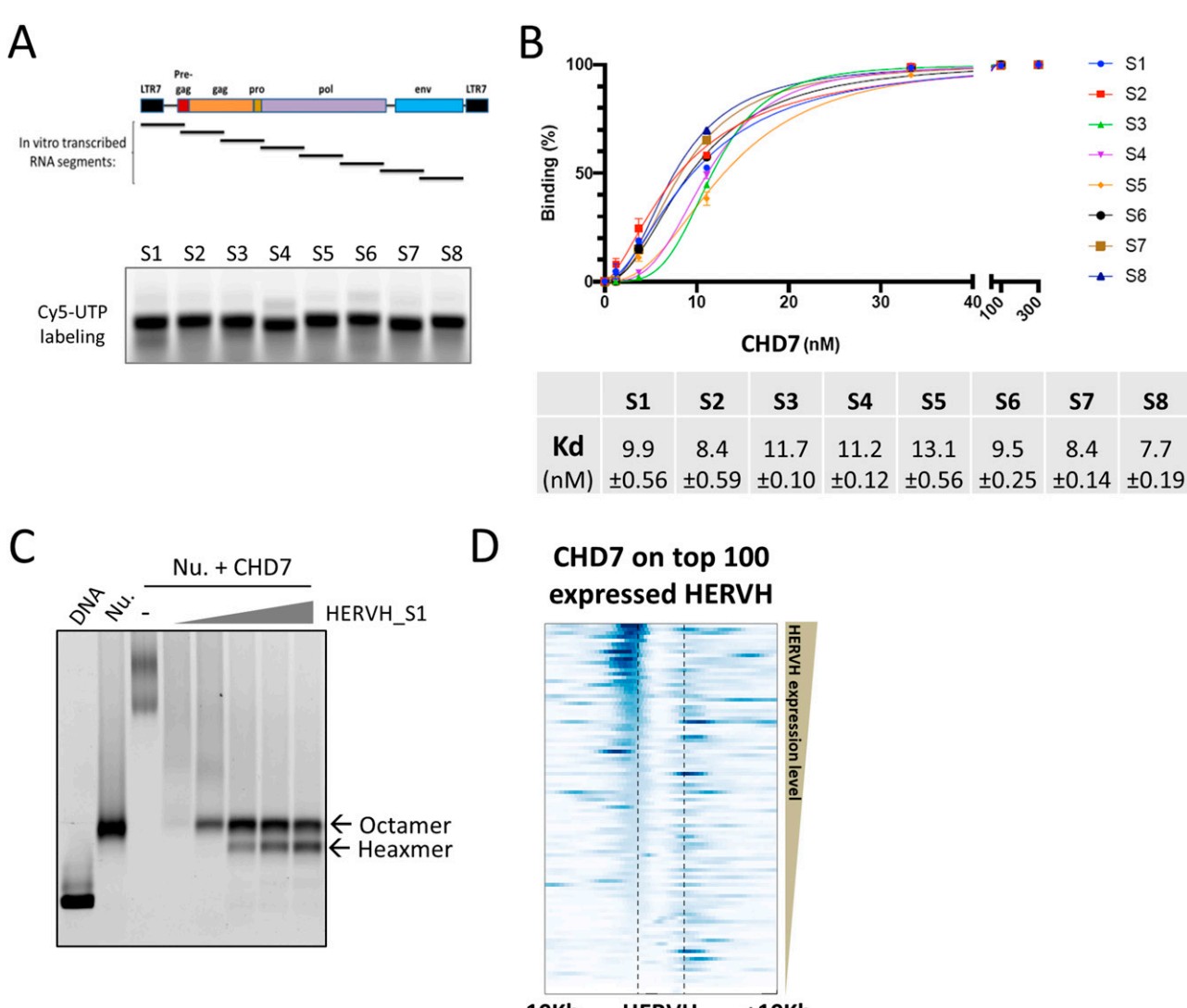

**Figure 5. HERVH RNA binds to CHD7 with high affinity and low specificity and disrupts interaction with a nucleosome.**
**(A)** Upper panel: schematic of eight segments of HERVH tested; lower panel: HERVH RNA segments were labeled with Cy-5 fluorescence and analyzed by agarose gel electrophoresis. **(B)** Quantification of binding between HERVH RNA segments and CHD7 using electrophoresis mobility shift assay. The Kd was calculated by fitting the data to the nonlinear specific binding with Hill coefficients. **(C)** HERVH RNA competed CHD7 away from nucleosome binding. The lower band after competition runs at the characteristic size for a hexamer. Concentrations of reagents in the reaction: 4 nM nucleosome; 500 nM CHD7; HERVH RNA segment-1 ranging 4–64 nM in twofold increments. **(D)** The top 100 expressed HERVH genes had strong binding of CHD7 on both sides of the repeat elements which was more pronounced on the 5′ end. Source data are available for this figure.

expressed HERVH genes have high levels of CHD7 immediately adjacent to the coding region, perhaps because of the presence of LTR sequences at each end of the HERVH repeat that might function as enhancers (Fig 5D). We found that the occupancy of CHD7 is stronger near the 5′ end for HERVH repeats with higher expression level than those with lower expression level (Fig 5D), as expected if these binding interactions helped to drive sense transcription. If binding of CHD7 to DNA is inhibited by interaction between HERVH and CHD7, we would predict that lowering HERVH levels using LNA knockdown would increase CHD7 binding at adjacent sites. We used LNA to knockdown HERVH and assessed the impact on CHD7 binding adjacent to HERVH repeats. We examined the binding of CHD7 on HERVH loci with expression levels greater than RPKM of

one and compared that binding between WT and HERVH knockdown cells. For HERVH elements where expression was reduced by LNA knockdown, we observed a small but detectable increase of CHD7 binding on regions adjacent to HERVH loci (Fig S6D. left panel). In contrast, CHD7 binding was not changed on HERVH elements where their expression was not affected by LNA (Fig S6D. right panel). These data are consistent with the possibility that local concentration of HERVH might impact HERVH-CHD7 interactions; however, the small changes observed preclude any conclusion.

Specificity in interaction might also be examined by finding specific small domains of CHD7 that are responsible for binding HERVH. CHD7 is a large protein (336 KD) with multiple modes of interacting with nucleic acid in its functional role in remodeling

chromatin. We examined three large fragments for binding that span the protein and were unable to identify a portion of the protein that might include a discrete RNA binding domain (Fig S6E and F).

### The binding of HERVH to CHD7 prevents chromatin remodeling

We tested the hypothesis that the interaction between HERVH RNA and CHD7 could impact the ability of CHD7 to remodel a nucleosome in an ATP-dependent manner. We measured nucleosome remodeling using a restriction enzyme accessibility assay. For substrate, we reconstituted a mononucleosome with Cy-5 labeled on a DNA template that could be remodeled by CHD7 in the presence of ATP to expose an MfeI restriction site for digestion (Fig 6A). The remodeling assay was carried out with the nucleosome (0.23 nM DNA concentration) incubated with 40 nM CHD7 protein. RNA was added to the reaction at concentrations ranging from 0.25 to 16 nM increasing by twofold increments. The presence of RNA did not interfere with the DNA cleavage activity of MfeI as evaluated at two concentrations of the enzyme (Fig S7A). In the presence of HERVH RNAs, the remodeling efficiency of CHD7 was diminished when the binding of RNA and protein was saturated (Fig 6B, exemplified with S1 segment; Fig S7B, full panel of all segments and quantification). In contrast, the presence of tRNA which does not strongly bind to CHD7 showed limited inhibition of remodeling even at high concentrations (Fig 6B). The presence of GFP RNA also inhibited remodeling (Fig S7C). These data were consistent with the electrophoresis mobility shift assay analyses (Figs 5B and S6C) in that there was no sequence specificity for inhibition of remodeling. We conclude that HERVH RNA binds CHD7 with high affinity but low sequence-dependent specificity and that this RNA–protein interaction inhibits the ability of CHD7 to remodel nucleosomes.

# Discussion

We have identified the HERVH family of lncRNAs as regulators of CHD7 function in pluripotent stem cells. We propose a model for regulation of CHD7 function by HERVH lncRNAs (Fig 6C): Normally in human ESCs, HERVH family members modulate binding of CHD7 to targeted enhancers to regulate enhancer function and generate expression patterns important for maintaining cells in a pluripotent state. This might involve the previously described interaction between CHD7 and p300 and modulation of levels of H3K27ac at enhancers (Schnetz et al, 2010). Consistent with this model, we found that acetylation of H3K27 on enhancers that normally had high levels of binding by CHD7 was diminished when CHD7 was knocked out. When HERVH lncRNAs were depleted, the binding of CHD7 to an important subset of enhancers was increased, with a corresponding increase in H3K27ac and apparent enhancer activity as measured by transcription from adjacent genes. The genes that were activated upon HERVH depletion included those in pathways involved in differentiation processes. We therefore propose that the presence of HERVH RNAs in pluripotent cells and the decreased level of this family of lncRNAs as cells differentiate play a key role in modulating gene expression during differentiation via interaction with CHD7.

Our data support previous proposals that CHD7 plays a key role in regulating the expression of genes needed for appropriate differentiation, which, when misregulated, might contribute to CHARGE syndrome when CHD7 function is impaired by mutation (Bajpai et al, 2010; Okuno et al, 2017). It is possible that these effects might be driven by interaction of CHD7 with the subset of enhancers we describe in this study. The observation that CHD7 binds to and regulates hundreds of enhancers for genes involved in differentiation raises the possibility that changes in levels of functional CHD7 in the developing embryo will lead to poorly modulated function of those enhancers and defects in appropriate levels of genes necessary for proper differentiation.

HERVH lncRNAs are highly expressed in hESCs and their expression level is closely correlated with cell pluripotency (Kelley & Rinn, 2012; Lu et al, 2014; Ohnuki et al, 2014; Wang et al, 2014a). The presence of HERVH lncRNAs negatively affect CHD7 binding to chromatin. However, the massive amount of HERVH RNA in hESCs does not prevent the binding of CHD7 to its target sites. The role for HERVH RNA in regulation of CHD7 binding therefore appears to be one of tuning, rather than an "all-or-none" dominant regulation of binding. When hESCs commit to differentiation, the expression of the HERVH family drops rapidly and the variance of HERVH expression can be greater than a 1,000-fold between pluripotent and differentiated states of cells (Lu et al, 2014; Ohnuki et al, 2014; Wang et al, 2014a). This immense change of expression level of HERVH family members during differentiation could trigger a reshaping of enhancer function genome wide with attendant changes in the transcriptome. We infer from our analysis that these changes occur, at least in part, via a CHD7-dependent mechanism. It would appear, given the large amount of HERVH lncRNAs in the cell and the high binding affinity of CHD7 for HERVH, and for other RNAs, that other factors in addition to HERVH combine to modulate CHD7 binding and function.

In addition to HERVH family lncRNAs, RNAs interacting with CHD7 identified from PAR-CLIP were transcribed from disparate parts of the genome and the majority of them (~60%) originated from protein coding regions (Fig S1C). Some of these RNAs might also regulate binding in a manner analogous to HERVH. Alternatively, as has been proposed previously for CBP associated RNAs, the CHD7 interacting RNAs might interact with CHD7 where they are transcribed and modulate expression from their gene (Bose et al, 2017). Thus, it is possible that interactions with numerous RNAs modulate CHD7 function genome wide, with HERVH lncRNAs playing a prominent role because of their abundant expression from a large number of the repeat sequences throughout the genome.

The strong enrichment for HERVH sequences in the PAR-CLIP analysis implies a significant degree of specificity in interactions between CHD7 and HERVH in the cell (Fig 1). Nevertheless, we were unable to identify any significant specificity for binding in vitro (Fig 5). We examined the possibility that spatial proximity between the HERVH RNA and CHD7 might generate specificity, a possibility raised by the observation that transcriptionally active HERVH genes have immediately adjacent CHD7 binding sites. We observed an extremely modest amount of increased binding by CHD7 to these proximal sites when we performed a HERVH knockdown, which is consistent with some degree of interaction between the newly transcribed HERVH and the adjacent CHD7 protein. However, the effects were so small that it would appear unlikely that this is a significant driver of specificity. Additional proteins might be

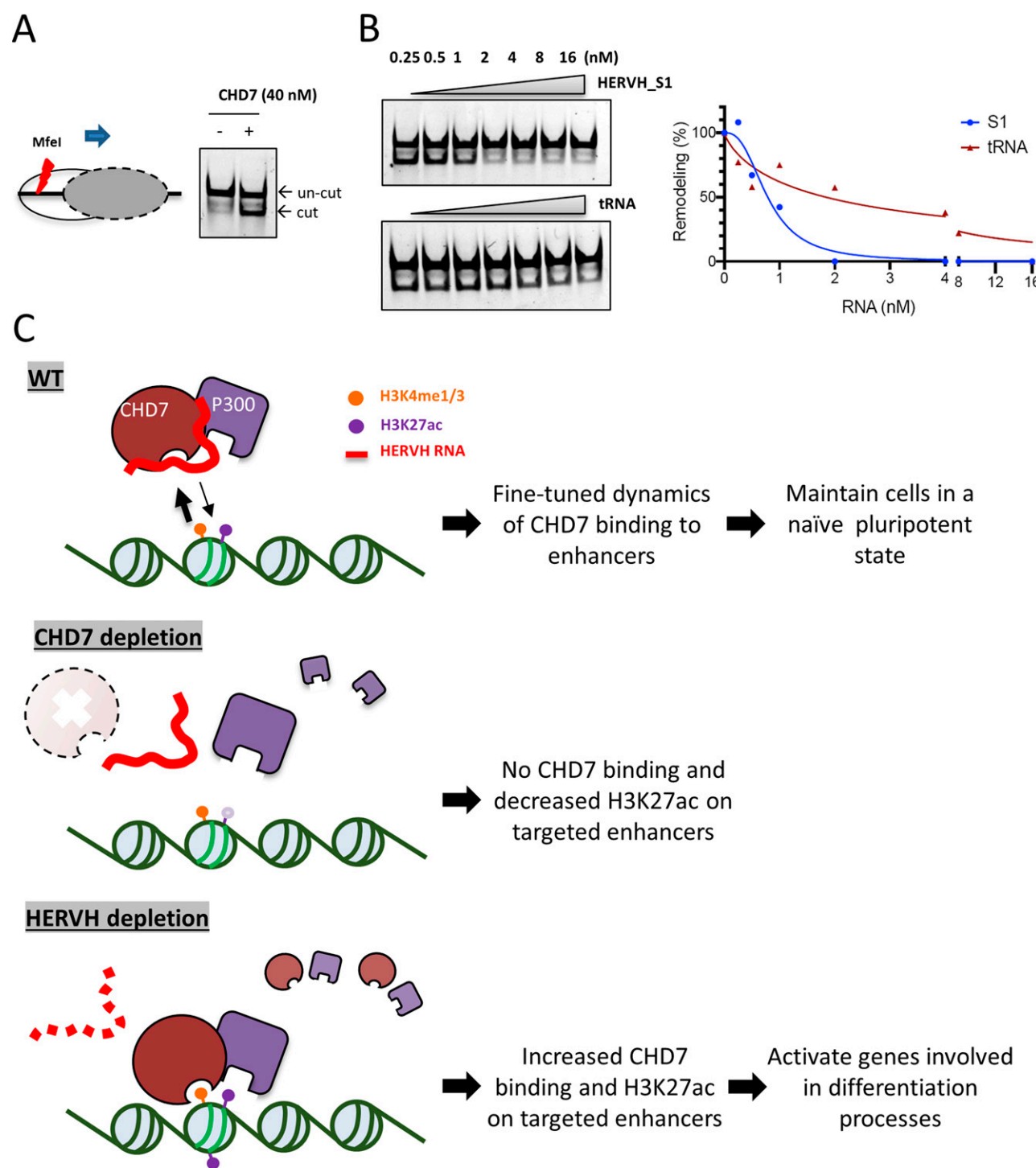

**Figure 6. The interaction between HERVH and CHD7 prevents chromatin remodeling mediated by CHD7.**
**(A)** Nucleosome remodeling was measured by a restriction enzyme accessibility protocol that measured the accessibility of an MfeI site located 28 base pairs from the entry point of the nucleosomal DNA. **(B)** HERVH RNA segment-1 and tRNA were introduced into a remodeling assay with increasing concentrations in twofold increments. Quantification of the amount of digested nucleosomal DNA is shown (right panel) with 100% defined as the amount of cleavage with no RNA added to the reaction. **(C)** A proposed model for HERVH and CHD7 interaction in regulating chromatin dynamics and gene expression. In WT cells, HERVH interacts with CHD7 and slows the dynamic of CHD7 binding and function at targeted enhancers. CHD7 function likely involves the recruitment of acetyltransferase p300 to CHD7 binding sites as H3K27ac levels are significantly reduced when CHD7 is absent. When HERVH is depleted, the binding and function of CHD7 at enhancers is intensified which leads to increased H3K27ac level and activation of genes involved in differentiation pathways.

required to generate binding specificity. P300 and mediator coactivator complexes have been shown to be associated with HERVH (Lu et al, 2014). It is possible that these or other, as yet uncharacterized, interactors might enhance the binding specificity between CHD7 and HERVH.

Accumulating evidence, including the study presented here, reveal the potential for repeat sequences interacting with partner proteins to impact chromatin dynamics and hence gene regulation. Almost half of the human genome is composed of repeat sequences and many of them are expressed in a spatial and time-specific manner. It is possible that lncRNAs transcribed from loci of these repeats may play indispensable roles in many cellular events in various cell lineages.

# Materials and Methods

## Cell culture

Human ES cell line H9 was cultured and maintained in mTesR1 medium (Stem Cell Technologies). For normal growth, the passaging procedure was performed as described previously (Beers et al, 2012). In brief, cells were washed twice with Dulbecco's phosphate-buffered saline (DPBS, Gibco) and incubated in DPBS containing 5 mM EDTA at RT for 5 min. The EDTA solution was aspirated and fresh mTesR1 was added to cells. Cells were detached from culture dish using gentle pipetting. Fresh culture plates or petri dishes were coated with Geltrex (Thermo Fisher Scientific) which was mixed in cold DMEM/F12 (1:1) medium (Gibco) with a ratio of 1:130 and solidified at 37°C for 1 h. Cells were seeded to the Geltrex coated well or plate supplemented with fresh mTesR1 medium and cultured in an incubator with 5% $CO_2$ at 37°C. Medium was changed every day. For cells used in further experiments, for example, gene expression knocked down by LNA-GapmeRs, TrypLE Express (Gibco) was used instead of EDTA solution which breaks clumped cells into single cells. In this case, 10 mM ROCK inhibitor (EMD Millipore) was added to the medium to facilitate single cell survival (Beers et al, 2012).

## PAR-CLIP

PAR-CLIP experiments were performed by following Hafner (2010a, 2010b) with some modifications. hESCs were cultured in mTeSR1 medium to ~80% confluency and nascently transcribed RNA was labeled by adding 4-thiouridin (Sigma-Aldrich) to the medium at 100 $\mu$M for 14 h. Cells were then washed twice with cold PBS in Petri dishes and were placed on ice with lids removed for UV-cross-linking. Cells were cross-linked with 0.15 J/cm$^2$ of 365 nm UV light in a Stratalinker 2400 (Stratagene), scraped from plates, and transferred to a new centrifuge tube. Cell pellets were collected at 700$g$ at 4°C for 5 min and the supernatant was removed carefully. Cell nuclei were extracted and sheared by Qsonica in the presence of RNasin Plus (to final 0.15 U/$\mu$l; Promega) and Sarkosyl (Sigma-Aldrich, to final 0.2%) with the program: on-20 s, off-40 s, amplitude- 45%, for a total on-time of 2 min. After sonication, samples were diluted with one volume of CLIP-IP buffer (10 mM Tris–HCl pH 7.4, 200 mM KOAc, 1 mM EDTA and 0.5 mM EGTA, 5% glycerol, 0.5%

Triton X-100, and 0.1% NP40), and 5 $\mu$l of RQ1 DNase (Promega)/400 $\mu$l sample was added to the samples and incubated at 37°C for 10 min then immediately transferred to ice. Samples were cleared by centrifugation at 17,000$g$ for 5 min at 4°C and the supernatants were carefully transferred to new tubes for immunoprecipitation. Dynal Magnetic beads (Invitrogen) were washed and equilibrated in CLIP-IP buffer before use. 5 $\mu$g of each IgG or CHD7 (ab31824; Abcam) antibodies were added to washed beads and incubated at 4°C for 2 h on a rotating platform. Conjugated beads-antibody was washed twice and resuspended in CLIP-IP buffer.

Sheared samples were added to freshly prepared antibody conjugated beads and rotated at 4°C for 2 h. The sample-bound beads were separated from solution by DynaMag-2 Magnet (Thermo Fisher Scientific) and washed twice with high-salt buffer (50 mM Tris–HCl, pH 7.4, 500 mM KCl, 0.05% NP40, and 1 mM EDTA), then twice with wash buffer (50 mM Tris–HCl, pH 7.4, 10 mM MgCl$_2$, and 0.2% Tween-20). The beads were resuspended in 45 $\mu$l dephosphorylation buffer (50 mM Tris–HCl, pH 7.4, 100 mM NaCl, 10 mM MgCl$_2$, and 1 mM DTT [add fresh]). CIP (New England Biolabs, to final 0.5 U/$\mu$l) and 0.5 $\mu$l of RNasin Plus were added to resuspended samples and incubated at 37°C for 10 min. Beads were washed twice with phosphatase wash buffer (50 mM Tris–HCl pH 7.4, 20 mM EGTA, 0.5% NP40) and twice with PNK buffer (50 mM Tris–HCl, pH 7.4, 50 mM NaCl, and 10 mM MgCl$_2$). Beads were resuspended in 45 $\mu$l of PNK buffer containing 5 mM freshly added DTT. [$\gamma$-32P]-ATP (PerkinElmer, to final 0.5 $\mu$Ci/$\mu$l) and T4 PNK (New England Biolabs, to final 1 U/$\mu$l) and 0.5 $\mu$l of RNasin Plus were added and samples were incubated at 37°C for 30 min. Samples were mixed manually every 5–10 min. 100 $\mu$M nonradioactive ATP was added to the solution and incubated at 37°C for another 5 min. Beads were washed five times with PNK buffer and resuspended in 30 $\mu$l of 1× SDS–PAGE sample loading buffer (Thermo Scientific Pierce) and incubate in a heat block at 90°C for 5 min to release the immunoprecipitated RNA binding protein with radiolabeled RNAs from the beads. The samples in solution were separated from beads on the separator and transferred to clean tubes. Samples were separated in a 4–12% NuPAGE Bis-Tris gel (Invitrogen) with 1× MOPS SDS running buffer (Invitrogen). Samples were transferred to a nitrocellulose membrane using a wet transfer apparatus. After transfer, the membrane was rinsed with PBS buffer, wrapped in clingfilm and exposed to a film for signal detection. The region containing RNA–protein complexes was identified by using the autoradiograph film and excised from the membrane to a new tube. This piece of membrane was cut into several small slices before adding 140 $\mu$l 1× PK buffer (200 mM Tris–HCl, pH 7.4, 100 mM NaCl, and 20 mM EDTA) and 10 $\mu$l proteinase K and incubated at 37°C for 20 min with 1,000 rpm shaking on a vortexer. An equal volume of urea buffer (100 mM Tris–HCl, pH 7.4, 50 mM NaCl, 10 mM EDTA, and 7 M urea [add fresh]) was added to the solution and incubated at 37°C for 20 min with 1,000 rpm shaking on a vortexer. RNAs were extracted by TRIzol reagent (Invitrogen) and cleaned up using Zymo RNA clean and concentrator-5 kit (Zymo Research) with DNase I treatment. RNA samples were recovered in nuclease-free water and converted to cDNA using Ovation RNA-Seq FFPE system kit (NuGEN). Sequencing libraries were prepared as described previously (Bowman et al, 2013).

## CUT&RUN

CUT&RUN was performed as described in Skene and Henikoff (2017). In brief, ~5 × 10$^5$ cells were used per reaction and were fixed with 1% formaldehyde at RT for 10 min. Cross-linking was terminated by

adding glycine at final concentration of 125 mM before subjecting the cells to the CUT&RUN protocol. After targeted fragments were released, 2 μl 10% SDS (to 0.1%), and 2.5 μl Proteinase K (20 mg/ml) were added to each sample and mixed by inversion and incubated at 65°C for 4 h up to O/N for de-crosslinking followed by phenol/chloroform extraction and ethanol precipitation. Samples were recovered in 25 μl 1 mM Tris–HCl, pH 8, and 0.1 mM EDTA and were kept at −20°C before sequencing library construction. Sequencing libraries were prepared as described previously (Bowman et al, 2013).

### Preparation of RNA samples for strand-specific sequencing

RNA was extracted from cells using TRI Reagent (Molecular Research Center, Inc) and further purified using Zymo RNA clean and concentrator-5 kit (Zymo Research) with DNase I treatment. The ribosomal RNAs were removed using NEBNext rRNA Depletion Kit v2 (New England Biolabs) before subjecting the samples to sequencing library construction. Strand-specific RNA sequencing libraries were prepared as described previously (Borodina et al, 2011).

### Bioinformatics analysis

PAR-CLIP sequencing reads were mapped to the human hg19 reference genome using BWA (Li & Durbin, 2010). PARalyzer peaks were called by PARalyzer v2.0 with default parameter of at least one conversion count per cluster (Corcoran et al, 2011). For the additional filtering by the magnitude of PAR-CLIP signal compared to input, we called peaks using SPP (Kharchenko et al, 2008). 100% of SPP peaks overlapped with PAR-CLIP peaks, comprising a more stringent subset of peaks, which were further filtered by the read density cutoff of at least 10 reads per kilobase per million reads (RPKM), resulting in two highly consistent sets of 3,821 and 2,598 PAR-CLIP peaks in two biological replicates. For CUT&RUN, paired end reads were trimmed using cutadapt and reads longer than 30 bp were used further for alignment. Reads were aligned to the human hg19 reference genome using bowtie2 and filtered using SAMtools (Li et al, 2009) to keep uniquely aligning reads and to remove PCR duplicates. CHD7 and HA peaks were called using Homer's findPeaks function for broad peaks. Only peaks that were >1 rpkm average (four replicates) in wild-type and >2FC average (four replicates) in knockout samples were used for further analyses. Bedtools intersect function was used to find overlap between peaks. Read densities within these peaks were quantile normalized within the sets of samples for the same antibody. Differential CUT&RUN read densities of H3K27ac and CHD7 at enhancers were identified using edgeR (Robinson et al, 2010) using the cutoffs of at least 1.5-fold difference and FDR < 0.01.

RNA-seq reads were mapped to the human hg19 reference genome with ENSEMBL annotation (GRCh37.75) using STAR aligner (Dobin et al, 2013). Read counts for individual transcripts were produced with HTSeq-count (Anders et al, 2015), followed by the estimation of expression values and detection of differentially expressed transcripts using EdgeR (Robinson et al, 2010). DEGs were defined by at least twofold change with FDR less than 0.01. To estimate the statistical significance of the association between DEGs and enhancers with differential H3K27ac density, we randomly shuffled genomic positions of these enhancers 1,000 times and compared the resulting random distribution of the number of DEGs falling within 100 kb of an enhancer to the number observed in our experiment, calculating Gaussian Z-score and P-value for this observed number.

Sequencing data generated in this study can be accessed in Gene Expression Omnibus (GEO): GSE171139 (See the Data Availability section). ChIP-seq of H3K4me3 and H3K36me3 used in this study were downloaded from GEO: GSM605316 and GSM605310, respectively.

### HERVH knockdown by LNA oligonucleotides

The design of LNA oligonucleotides was based on the study by Wang et al (2014a) in which three sequences of shRNAs (#3, #4 and #12) targeting HERVH were shown to have effective knockdown efficiency (Wang et al, 2014a). HERVH LNA oligonucleotides were ordered from Exiqon (QIAGEN). For each well of a 24-well plate, $1 \times 10^5$ cells were seeded on Geltrex coating with 1 ml mTesR1 medium containing 10 μM RCOK inhibitor. Cells were placed in an incubator with 5% $CO_2$ at 37°C for 2–4 h to allow them to re-attach to the plate before transfection. LNA-GapmeRs were prepared with nuclease-free water at 50 μM stock concentration. Lipofectamine Stem Reagent (Invitrogen) was warmed to reach room temperature before use. Per reaction, 100 pmol in total of LNA oligonucleotide(s) and 3.5 μl of Lipofectamine Stem Reagent were added to 100 μl Opti-MEM. The LNA and Lipofectamine Stem Reagent containing tubes were mixed and incubated at RT for 10 min. LNA-Lipofectamine Stem Reagent mixture was added to seeded cells with fresh mTesR1 medium. The medium was gently mixed and cells were returned to the incubator. Cells were harvested after 24-h treatment either for RNA extraction or for CUT&RUN protocols.

### Generation of cell lines by CRISPR-Cas9

Genome editing in human pluripotent stem cells was achieved by CRISPR-Cas9 as described previously (Kim et al, 2014; Liang et al, 2015). In brief, CRISPR guide RNAs with the lowest off-target score were identified using CRISPR Design software (crispr.mit.edu). The selected CRISPR RNAs (CrRNAs) and universal tracrRNA were provided by Integrated DNA Technologies, Inc. (IDT). Cas9 protein (5 mg/ml) was prepared by our own laboratory. For homology-directed repair, 200 nucleotide single-stranded oligo donor (ssODN) templates with roughly equal length of homology arms from the mutation site were provided by IDT. Universal tracrRNA and crRNA were prepared at 200 and 100 μM in RNase-free duplex buffer, respectively. 2.5 μl of the tracrRNA was added to the reconstituted 5 μl crRNA, gently vortexed, placed at 95°C and gradually cooled down to RT to allow hybridization. For CRISPR/Cas9 RNP assembly, 7.5 μl of hybridized crRNA/tracrRNA (133.3 pmol) was added to 50 μg of Cas9 protein (312 pmol), mixed gently and incubated at RT for 25 min. Upon completion of RNP complex assembly, 3 μl of 50 ng/μl linear puromycin (Clontech) was added to the RNP assembly. 2 μl of 50 μM ssODN/homology-directed repair template was added when appropriate. While the RNP was being assembled, hESCs were trypsinized by TrypLE and $8 \times 10^5$ hESCs were

resuspended in 100–110 $\mu$l of nucleofection reagent (Lonza). The RNP was mixed with resuspended cells and transferred to an electroporation cuvette. The cuvette was placed in an electroporation machine (Lonza Amaxa) and run with program B-16. 1 ml of warm mTesR1 medium containing 10 $\mu$M ROCK inhibitor was added to cells after transfection. Cells were then transferred and distributed equally to two 10-cm plates that were coated with Geltrex and contained 9.5 ml of mTesR1 medium with 10 $\mu$M ROCK inhibitor. Cells were cultured in an incubator with 5% $CO_2$ at 37°C for 24 h before starting the puromycin selection at 1 $\mu$g/ml for 48 h. Cells were cultured for another 7–10 d in the medium without puromycin. The medium was changed daily. After cell growth, individual colonies were picked, expanded, and screened by PCR with designated primer sets for successfully engineered cells.

## Western blot

Cells were washed twice with cold PBS prior being lysed with RIPA buffer (25 mM Tris–HCl pH 8, 150 mM NaCl, 1% NP-40, 0.5% sodium deoxycholate, 0.1% SDS, 1× protease inhibitor [cOmplete, EDTA-free, add freshly; Roche]) on ice for 20 min. Benzonase was added to the cell lysate to help remove genomic DNA. The cell lysate was centrifuged at 14,000x$g$ for 15 min to pellet the cell debris. The supernatant was carefully transferred to a new tube and the protein concentration was measured by Bradford reagent (Bio-Rad). 25 $\mu$g total protein per sample was mixed in 1× SDS–PAGE sample loading buffer and incubated at 95°C for 10 min. Samples were separated in a 4–12% NuPAGE Bis-Tris gel with 1× MOPS SDS buffer and then transferred to a nitrocellulose membrane. The membrane was incubated in TBST containing 5% milk at RT for 1 h for blocking and then hybridized with primary antibody at 4°C for overnight. The next day the membrane was washed three times with TBST at RT for 10 min and hybridized with HRP-conjugated secondary antibody at RT for 1 h. The membrane was washed again three time with TBST at RT for 10 min before subjecting to Clarity Western ECL solutions (Bio-Rad) for signal detection by Amersham Hyperfilm (GE Healthcare).

## Protein expression and purification

The construction and purification of full-length double-tag CHD7 and CHD7 mutants have been described (Bouazoune & Kingston, 2012). In brief, proteins only bearing a FLAG-tag were purified with M2-affinity gel (Invitrogen). Dual-tagged full-length CHD7 protein was purified first with Ni-NTA resin (QIAGEN) and subsequently with M2-affinity gel.

## Nucleosome preparation

The "601" nucleosome-positioning DNA sequence was used for nucleosomal template preparation (Lowary & Widom, 1998). The DNA template was amplified by PCR and fluorescently labeled using a Cy-5 conjugated reverse primer. The PCR product was purified with a PCR purification kit (QIAGEN) before nucleosome reconstitution. For nucleosome assembly, the DNA template was mixed with recombinant histone octamer in a 1:3 M ratio and a twofold DNA mass of sheared salmon sperm DNA (Ambion) and the mixture was then dialyzed in dialysis buffers (10 mM Tris–HCl, pH 7.5, 0.1% NP-40, 0.2 mM EDTA, and 5 mM 2-mercaptoethanol) containing various salt concentrations (2, 1.5, 1, 0.75, 0.5 M, and 10 mM NaCl, respectively) from high to low salt stepwise as described previously (Hsieh et al, 2015).

## Cloning of HERVH segments

Full-length HERVH on Chromosome 8: 132,322,422–132,328,094 was first amplified from the genome by PCR with primers targeting the unique sequences outside of the repeat element using Phusion High-Fidelity PCR Master Mix reagents (New England Biolabs). The full-length PCR fragment was then used as template for amplification of eight divided HERVH segments (~0.8 kb of each) by PCR with designated primer sets. The amplified PCR products were subcloned into pCR4 Blunt-TOPO vector (Thermo Fisher Scientific) and verified by Sanger sequencing.

## HERVH RNA preparation by in vitro transcription

Individual HERVH segments were amplified with T7 promoter containing forward primer and specific reverse primers by PCR and purified using a PCR purification kit (QIAGEN) and used as templates for in vitro transcription. The corresponding RNAs were prepared with Cyanine 5 (Cy5)-UTP labeling by T7 MEGAscript kit (Thermo Fisher Scientific).

## Electrophoresis mobility gel shift assay

RNA in nuclease-free water was heated at 95°C for 1 min and cooled on ice for 2 min before adding to 1× Binding Buffer (20 mM Hepes, pH 7.9, 10% glycerol, 150 mM KCl, 2.5 mM $MgCl_2$, 1 mM DTT [add freshly], 0.1 mg/ml BSA [add freshly], and RNasin Plus [to final 0.1 U/$\mu$l, add freshly]). RNA was refolded at 37°C for 30 min. Serial dilutions of protein were prepared with 1× Binding Buffer. After refolding, RNA (final 1 nM) was added to protein solutions. The reactions were well mixed and incubate at RT for 30 min. Orange-G dye (to final 0.04%) was added and mixed with samples before subjecting to 0.7% Seakem Gold agarose gel (Lonza) for analysis. Electrophoresis was performed at 400 V for 10 min and then at 200 V for 2 h in 0.5× Tris-borate-EDTA (TBE) buffer. Gel images were acquired using a Typhoon PhosphorImager and analyzed by ImageJ software. Gels were analyzed and quantified by ImageJ. The dissociation constant (Kd) was calculated by fitting the data to the nonlinear specific binding with Hill slope.

## Competition assay

RNA in nuclease-free water was heated at 95°C for 1 min and cooled on ice for 2 min before adding to 1× Binding Buffer (20 mM Hepes pH 7.9, 10% glycerol, 150 mM KCl, 2.5 mM $MgCl_2$, 1 mM DTT [add freshly], 0.1 mg/ml BSA [add freshly], and RNasin Plus [to final 0.1 U/$\mu$l, add freshly]). RNA was refolded at 37°C for 30 min. After refolding, serial dilutions of RNA were prepared with 1× Binding Buffer. A nucleosome preparation with Cy-5 labeled on a DNA template and CHD7 protein was mixed in 1× Binding Buffer and incubated at RT for 10 min. Folded RNA with various concentrations was then added to the pre-mixed nucleosome-protein solution and incubated at RT for 20 min. The final concentration of CHD7 was 500 nM and the final concentrations of RNA were ranging from 4 to 64 nM in twofold increment in reactions. Orange-G dye (to final 0.04%) was added and mixed with samples before subjecting to 0.7% Seakem Gold agarose gel (Lonza) for analysis. Electrophoresis was at 250 V for

5 min and then at 150 V for 1 h 30 min in 0.5× TBE buffer. Gel images ware acquired using a Typhoon PhosphorImager and analyzed by ImageJ software.

### Restriction enzyme-accessibility assay

The restriction enzyme accessibility assay was performed as described previously with modifications (Grau et al, 2011). In brief, RNA in nuclease-free water was heated at 95°C for 1 min and cooled on ice for 2 min before adding to 1× CutSmart Buffer (20 mM Tris-acetate, pH 7.9, 50 mM potassium acetate, 10 mM magnesium acetate, and 0.1 mg/ml BSA) and was refolded at 37°C for 30 min. After refolding, serial dilutions of RNA ranging from 0.25 to 16 nM in twofold increment of concentration were prepared with 1× CutSmart Buffer and mixed with CHD7 (40 nM) and incubated at RT for 20 min. The nucleosome preparation with Cy-5 labeled on DNA template was then added to the RNA-protein mixture and incubated in the presence of 2 mM ATP and 1.5 U/$\mu$l of MfeI-HF (New England Biolabs) at 30°C for an hour. The remodeling reaction was stopped by adding 1× Stop Buffer (4 mM Tris–HCl, pH 7.5, 25 mM EDTA, 0.4% SDS, 0.02% Orange G, and 8% glycerol) and then mixed well with Proteinase K (to final 1 mg/ml) and incubated at 55°C for 45 min. The remodeling was measured by the ability of remodeling factors to expose an MfeI restriction site for digestion at +28-bp of the nucleosomes. The digested samples were separated by 6% native PAGE gel in 1× TBE buffer. The images were taken by Amersham Typhoon (GE Healthcare Life Sciences) and analyzed using ImageJ software. The remodeling efficiency was quantified and presented by fitting the data to the non-linear specific binding with Hill slope.

### List of expression, effect of knockdown, and PAR-CLIP detection for all annotated HERVH genes

Please see Table S1. For all HERVH transcripts annotated in the human genome, the name, genomic location, RNA-seq expression levels (RPKM) in wild type and HERVH knockdown cells, and expression change upon HERVH knockdown ($\log_2$ fold change) are indicated, as well as the overlap with a CHD7 PAR-CLIP peak (YES/NO), the name of the closest other gene, and the distance to this gene.

### List of antibodies

Please see Table S2.

### List of primers

Please see Table S3.

### List of guide RNAs and templates for homologous recombination

Please see Table S4.

### List of LNA-GapmeRs

Please see Table S5.

## Data Availability

The sequencing data from this publication have been deposited to the GEO database (https://www.ncbi.nlm.nih.gov/geo/) and assigned the identifier GSE171139.

## Supplementary Information

## Acknowledgements

We thank J Kim for the help with immunostaining experiments, J Kim and C Tsokos for critical reading of the manuscript, and the RE Kingston laboratory for fruitful discussions. This work was supported by the National Institutes of Health R01GM04390 and R35GM131743 to RE Kingston.

### Author Contributions

F-K Hsieh: conceptualization, data curation, formal analysis, investigation, and writing—original draft, review, and editing.
F Ji: data curation, software, formal analysis, and writing—review and editing.
M Damle: formal analysis, validation, and writing—review and editing.
RI Sadreyev: data curation, supervision, and writing—review and editing.
RE Kingston: conceptualization, supervision, funding acquisition, investigation, and writing—original draft, review, and editing.

### Conflict of Interest Statement

The authors declare that they have no conflict of interest.

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
