## [Reviewer comments · Life Science Alliance]

Life Science Alliance

HERVH-derived lncRNAs negatively regulate chromatin targeting and remodeling mediated by CHD7

Fu-Kai Hsieh, Fei Ji, Manashree Damle, Ruslan Sadreyev, and Robert Kingston

DOI: <https://doi.org/10.26508/lsa.202101127>

Corresponding author(s): Robert Kingston, Massachusetts General Hospital

Review Timeline:

Submission Date:	2021-06-01
Editorial Decision:	2021-06-15
Revision Received:	2021-09-14
Editorial Decision:	2021-09-17
Revision Received:	2021-10-05
Accepted:	2021-10-06

Scientific Editor: Novella Guidi

Transaction Report:

Please note that the manuscript was previously reviewed at another journal and the reports were taken into account in the decision-making process at *Life Science Alliance*. Since the original reviews are not subject to Life Science Alliance's transparent review process policy, the reports and author response cannot be published.

June 14, 2021

Re: Life Science Alliance manuscript #LSA-2021-01127-T

Robert E Kingston
Massachusetts General Hospital
Molecular Biology
185 Cambridge Street
Simches 7th floor
Boston, MA 2114

Dear Dr. Kingston,

Thank you for submitting your manuscript entitled "The lncRNA HERVH negatively regulates chromatin targeting and remodeling mediated by CHD7" to Life Science Alliance. We invite you to submit a revised manuscript addressing the Reviewer comments.

Thank you for this interesting contribution to Life Science Alliance. We are looking forward to receiving your revised manuscript.

Sincerely,

Novella Guidi
Scientific Editor
Life Science Alliance

- A letter addressing the reviewers' comments point by point.
- An editable version of the final text (.DOC or .DOCX) is needed for copyediting (no PDFs).
- High-resolution figure, supplementary figure and video files uploaded as individual files: See our detailed guidelines for preparing your production-ready images, <https://www.life-science-alliance.org/authors>
- Summary blurb (enter in submission system): A short text summarizing in a single sentence the study (max. 200 characters including spaces). This text is used in conjunction with the titles of papers, hence should be informative and complementary to the title and running title. It should describe the context and significance of the findings for a general readership; it should be written in the present tense and refer to the work in the third person. Author names should not be mentioned.

B. MANUSCRIPT ORGANIZATION AND FORMATTING:

September 17, 2021

RE: Life Science Alliance Manuscript #LSA-2021-01127-TR

Dr. Robert E Kingston
Massachusetts General Hospital
Molecular Biology
185 Cambridge Street
Simches 7th floor
Boston, MA 02114

Dear Dr. Kingston,

Thank you for submitting your revised manuscript entitled "HERVH-derived lncRNAs negatively regulate chromatin targeting and remodeling mediated by CHD7". We would be happy to publish your paper in Life Science Alliance pending final revisions necessary to meet our formatting guidelines.

-please add the Twitter handle of your host institute/organization as well as your own or/and one of the authors in our system

Figures check:

-please indicate molecular weights for the protein blots in figures

-there is a possible horizontal splice in the first lane in figure 5C. Please provide source data for this figure.

-please provide scale bars for figure S2A

A. FINAL FILES:

B. MANUSCRIPT ORGANIZATION AND FORMATTING:

Sincerely,

October 6, 2021

RE: Life Science Alliance Manuscript #LSA-2021-01127-TRR

Dr. Robert E Kingston
Massachusetts General Hospital
Molecular Biology
185 Cambridge Street
Simches 7th floor
Boston, MA 02114

Dear Dr. Kingston,

Thank you for submitting your Research Article entitled "HERVH-derived lncRNAs negatively regulate chromatin targeting and remodeling mediated by CHD7". It is a pleasure to let you know that your manuscript is now accepted for publication in Life Science Alliance. Congratulations on this interesting work.

DISTRIBUTION OF MATERIALS:

Again, congratulations on a very nice paper. I hope you found the review process to be constructive and are pleased with how the manuscript was handled editorially. We look forward to future exciting submissions from your lab.

Sincerely,
